# Minimally Invasive Surgery: Standard of Care for Mitral Valve Endocarditis

**DOI:** 10.3390/medicina59081435

**Published:** 2023-08-08

**Authors:** Cristina Barbero, Marco Pocar, Dario Brenna, Barbara Parrella, Sara Baldarelli, Valentina Aloi, Andrea Costamagna, Anna Chiara Trompeo, Alessandro Vairo, Gianluca Alunni, Stefano Salizzoni, Mauro Rinaldi

**Affiliations:** 1Division of Cardiac Surgery, Cardiovascular and Thoracic Department, “Città della Salute e della Scienza di Torino”, Molinette Hospital, Corso Dogliotti 14, 10126 Turin, Italy; dbrenna89@gmail.com (D.B.); barbara.parrella@unito.it (B.P.); sara.baldarelli91@gmail.com (S.B.); valentina.aloi@unito.it (V.A.); stefano.salizzoni@unito.it (S.S.); mauro.rinaldi@unito.it (M.R.); 2Department of Surgical Sciences, University of Turin, 10126 Turin, Italy; andrea.costamagna@unito.it; 3Department of Clinical Sciences and Community Health (DISCCO), University of Milan, 20122 Milan, Italy; 4Division of Cardiac Intensive Care, Anesthesia, Intensive Care and Emergency Department, “Città della Salute e della Scienza di Torino”, Molinette Hospital, 10126 Turin, Italy; atrompeo@cittadellasalute.to.it; 5Unit of Echocardiography, Division of Cardiology, Cardiovascular and Thoracic Department, “Città della Salute e della Scienza di Torino”, Molinette Hospital, 10126 Turin, Italy; vairo.alessandro@gmail.com (A.V.); a.gianluca1@virgilio.it (G.A.)

**Keywords:** mitral valve, infective endocarditis, minimally invasive cardiac surgery, mini-thoracotomy, cardiac reoperation, prosthetic valve endocarditis, systemic embolization

## Abstract

*Background*. Minimally invasive surgery via right mini-thoracotomy has become the standard of care for the treatment of mitral valve disease worldwide, particularly at high-volume centers. In recent years, the spectrum of indications has progressively shifted and extended to fragile and higher-risk patients, also addressing more complex mitral valve disease and ultimately including patients with native or prosthetic infective endocarditis. The rationale for the adoption of the minimally invasive approach is to minimize surgical trauma, promote an earlier postoperative recovery, and reduce the incidence of surgical wound infection and other nosocomial infections. The aim of this retrospective observational study is to evaluate the effectiveness and the early and late outcome in patients undergoing minimally invasive surgery for mitral valve infective endocarditis. *Methods*. Prospectively collected data regarding minimally invasive surgery in patients with mitral valve infective endocarditis were entered into a dedicated database for the period between January 2007 and December 2022 and retrospectively analyzed. All comers during the study period underwent a preoperative evaluation based on their clinical history and anatomy for the allocation to the most appropriate surgical strategy. The selection of the mini-thoracotomy approach was primarily driven by a thorough transthoracic and especially transesophageal echocardiographic evaluation, coupled with total body and vascular imaging. *Results*. During the study period, 92 patients underwent right mini-thoracotomy to treat native (80/92, 87%) or prosthetic (12/92, 13%) mitral valve endocarditis at our institution, representing 5% of the patients undergoing minimally invasive mitral surgery. Twenty-six (28%) patients had undergone previous cardiac operations, whereas 18 (20%) presented preoperatively with complications related to endocarditis, most commonly systemic embolization. Sixty-nine and twenty-three patients, respectively, underwent early surgery (75%) or were operated on after the completion of the targeted antibiotic treatment (25%). A conservative procedure was feasible in 16/80 (20%) patients with native valve endocarditis. Conversion to standard sternotomy was necessary in a single case (1.1%). No cases of intraoperative iatrogenic aortic dissection were reported. Four patients died perioperatively, accounting for a thirty-day mortality of 4.4%. The causes of death were refractory heart or multiorgan failure and/or septic shock. A new onset stroke was observed postoperatively in one case (1.1%). Overall actuarial survival rate at 1 and 5 years after operation was 90.8% and 80.4%, whereas freedom from mitral valve reoperation at 1 and 5 years was 96.3% and 93.2%, respectively. *Conclusions*. This present study shows good early and long-term results in higher-risk patients undergoing minimally invasive surgery for mitral valve infective endocarditis. Total body, vascular, and echocardiographic screening represent the key points to select the optimal approach and allow for the extension of indications for minimally invasive surgery to sicker patients, including active endocarditis and sepsis.

## 1. Introduction

Minimally invasive surgery has become the standard of care for the treatment of mitral valve (MV) disease at many institutions worldwide. Compared to traditional median full sternotomy, equivalence in terms of the safety, efficacy, and durability of operations carried out via the right mini-thoracotomy has been widely documented with increasing favor for this approach by the cardiac surgical community. Non-inferiority of the former aspects is coupled with a better outcome in terms of postoperative pain, a lower rate of transfusions and reoperation for bleeding, improved cosmesis, shorter hospital length of stay and time to return to normal activity or work, and a lower need for rehabilitation resources at discharge [1,2,3,4,5,6]. As a consequence, especially at high-volume institutions, the spectrum of indications has progressively shifted toward more fragile and higher-risk patients allowing them to apply the minimally invasive approach to MV disease of increasing complexity, ultimately including patients with MV infective endocarditis (IE).

Surgical treatment in patients with IE is required in more than 50% of cases to prevent severe complications, most typically heart failure, uncontrolled infection, and systemic embolism [7,8]. US and European guidelines strongly recommend early intervention when surgery is needed. The evolution in the planning of an earlier surgical procedure, coupled with advances in intensive care and postoperative management have dramatically improved outcomes in these patients. However, the rates of perioperative mortality and morbidity are still consistently high [7,8].

The rationale for the adoption of the minimally invasive approach in these higher-risk patients is to minimize the surgical trauma, thus promoting a quicker recovery after the operation and reducing the rate of surgical wound infection and other hospital-related infections.

The aim of this retrospective observational study is to evaluate the effectiveness and the early and late outcome in patients undergoing minimally invasive MV surgery for IE.

## 2. Methods

### 2.1. Study Design and Patients’ Selection

Data regarding minimally invasive surgery in patients with MV IE were prospectively entered into a dedicated database for the period between January 2007 and December 2022. All patients described in our series met the modified Duke criteria for IE [9]. All comers during the study period underwent a preoperative evaluation based on clinical history and anatomy for the allocation to the most appropriate surgical setting.

Selection for the mini-thoracotomy approach was primarily driven by a thorough transthoracic and especially transesophageal echocardiographic evaluation. Although less invasive approaches have been extended to treat multiple left-sided valve diseases in recent years, any suspicion of even minimal aortic valve involvement by the IE mandates a contraindication for the minimally invasive approach at our institution [10,11]. Hence, in this clinical scenario, patients were scheduled for conventional full sternotomy to allow for, if indicated, the treatment of associated lesions and extensive inspection of the aortic valve and aorto-mitral curtain. Moreover, more than mild aortic valve regurgitation and severe or extreme pleural adhesions of the right lung following prior thoracic operations are to be considered major if not absolute exclusion criteria for the minimally invasive approach, irrespective of IE. Finally, patients operated on an emergency basis with cardiogenic shock were addressed via median sternotomy.

In case of previous cardiac surgical procedures, the endoaortic balloon setting was used in most cases to avoid tedious and near-blind dissection of adhesions to free the ascending aorta for cross-clamping. However, this was not always feasible, most typically in the case of a dilated ascending aorta, i.e., with a maximum diameter above 40 mm, or in the presence of moderate or severe tortuosity of the infrarenal aorta and ileofemoral axes. A small caliber of the common femoral vessels (<7 mm) also renders impractical the use of the endoballon, which requires the insertion of a 21F of 23F arterial perfusion cannula with a side branch for the insertion of the balloon itself, implying in these cases transthoracic clamping with a Chitwood or, more recently, a Cygnet type aortic clamp (see below). Moreover, in case of severe tortuous and/or atheromatous abdominal aortoiliac and femoral vessels, the antegrade perfusion via the axillary artery with transthoracic clamping was predominantly preferred to minimize the hazards of systemic embolization and stroke related to retrograde perfusion. In this respect, a rigorous peripheral vascular screening with a computerized tomography scan or aortography and ileofemoral angiography or a combination of imaging techniques became routine in our clinical practice since 2009 [12].

The selection of one setting with respect to the others was patient-oriented and independent of the learning curve. Long-term outcomes were obtained from regular postoperative follow up at the outpatient clinic, telephone interviews, or both. Early surgery relates to the course of IE and, more specifically, to operations carried out at any time during the course of antibiotic treatment [11]. Postoperative stroke was defined as clinical signs persisting at the time of discharge from the hospital and/or in the presence of localized ischemic infarcts detectable using conventional neuroimaging techniques. In relation to late follow up, adverse events analyzed as primary outcomes in this present study were death from any cause and MV reoperation. The cause of reoperation, i.e., recurrent IE or not, was also analyzed as a secondary outcome.

### 2.2. Surgical Technique

The right mini-thoracotomy approach, perfusion strategies, and aortic clamping techniques used for patients undergoing MV surgery have been described previously [12,13,14]. Briefly, a right anterolateral mini-thoracotomy in the fourth intercostal space is performed, and double lumen endotracheal tube intubation for the single left lung ventilation is provided in all patients. More posteriorly, a secondary port is prepared for the endoscope and for carbon dioxide insufflation. An additional sixth intercostal space port is created for pump suction. After the institution of cardiopulmonary bypass, the core temperature is lowered to 30 °C. Arterial perfusion is gained with peripheral femoral or axillary cannulation. The latter is usually preferred to provide antegrade systemic perfusion in the case of severe atherosclerotic burden [12]. Venous drainage is obtained via double femoral and jugular cannulation. All cannulae are inserted with a Seldinger technique, either under direct vision in case of vascular surgical exposure or percutaneously. The ascending aorta is clamped using the endoaortic balloon or a trans-thoracic clamp. In the endoaortic clamping setting, aortic occlusion and cardioplegia delivery are gained with a balloon catheter inserted into the sidearm of a femoral arterial cannula (21F or 23F Intraclude, Edwards Lifesciences, Irvine, CA, USA). In the trans-thoracic clamping setting, the clamp is addressed towards the ascending aorta using the first intercostal space with a Chitwood clamp or the main port with a Cygnet flexible clamp. Cardioplegia is delivered with a 7F cardioplegia needle (CalMed Technologies, Santa Inez, CA, USA) placed into the proximal ascending aorta. Antegrade myocardial protection is provided with St. Thomas (Plegisol, Hospira Inc., Lake Forest, IL, USA) or Custodiol (Bretschneider histidine, tryptophan, ketoglutarate solution, Kohler Chemie, Bensheim, Germany) cold crystalloid cardioplegia [13]. Superior and inferior vena cava snaring is obtained by placing tourniquets around the vessels or by placing endovascular balloons to provide a temporary mini right atriotomy to drain the cardioplegic solution and in patients requiring associated right atrial procedures, most commonly tricuspid valve repair. The MV is exposed via a standard left atriotomy parallel and posterior to the interatrial septum (Figure 1). The extension of vegetation, leaflet involvement, and the chances of valve repair are assessed. Clamp release is obtained at a core temperature above 33 °C during rewarming. Intraoperative transesophageal echocardiography is mandatory and was used in all patients to guide the correct positioning of the cannulae before the onset of cardiopulmonary bypass and to assess cardiac function, residual MV regurgitation in case of MV repair, paravalvular leaks, and prosthetic valve gradients after the intracardiac phase of the operation.

### 2.3. Statistical Analysis

Categorical and continuous, either normally distributed or skewed, variables are reported as n (%), mean ± standard deviation, or median [interquartile range] as appropriate. The probability of adverse events over time was analyzed with the Kaplan–Meier method. Factors affecting a different probability of an adverse event were assessed with the log-rank test. *p* values < 0.05 were considered statistically significant. SPSS software package was used for statistical computations.

## 3. Results

During the study period, 1847 patients underwent minimally invasive MV surgery at our institution. In 92 patients (5%), the diagnosis and indication for operation was IE. Among these, 80/92 (87%) were native MV IE, whereas the remaining 12/92 (13%) were prosthetic valve IE. Patient characteristics are summarized in Table 1. The mean age was 61.1 ± 12.7 years, and 34 patients were female (37%). Twenty-six patients (28%) had undergone one or more previous cardiac operations. Eighteen patients (20%) reported preoperative IE-related complications, namely, systemic embolic events in 13 cases and heart failure in 5 cases. Sixty-nine (75%) patients underwent early MV surgery, whereas thirteen (25%) patients were operated on after the completion of antibiotic treatment. Overall, the estimated operative mortality applying the logistic EuroSCORE was 14.9%. Micro-organisms were identified on excised tissues at the time of surgery in 71 (77%) patients and were almost invariably Gram-positive bacteria (Table 2). The morphology of vegetations was described inconstantly on echocardiograms and operative reports and could not be retrieved in all patients.

Intraoperative and postoperative variables are reported in Table 3. Isolated MV surgery was performed in most of the cases (82/92, 89%). Although IE most often precludes or renders MV repair unwise, a conservative procedure was possible in 16/80 (20%) patients with IE on the native MV. Retrograde systemic arterial perfusion via the femoral artery was performed in the vast majority of cases (85/92, 92%), whereas aortic cross-clamping was obtained with the endoluminar balloon in 40/92 (43%) and with a trans-thoracic clamp in 51 (55%) cases, respectively. The procedure was carried out without cardioplegic arrest on the fibrillating heart in one patient with multiple prior operations. Conversion to standard full sternotomy was necessary in a single patient (1.1%). No cases of iatrogenic intraoperative aortic dissection were observed.

Four patients died perioperatively, accounting for an operative thirty-day mortality of 4.4%. The causes of death were worsening heart or multiorgan failure, refractory sepsis, or a combination of these conditions. Postoperative stroke was reported in a single case (1.1%).

Clinical follow up was concluded in April 2023 and was 100% completed. The median follow up was 98.1 months. Overall, the Kaplan–Meier survival analysis outlined that an actuarial survival rate at 1 and 5 years was 90.8% and 80.4%, respectively (Figure 2). Eight patients underwent redo MV surgery during follow up. Surgical MV replacement was performed in 7 cases, whereas 1 patient received transcatheter valve-in-valve implantation to address bioprosthetic structural deterioration. Despite being an interventional procedure, the latter was considered an adverse MV event and was included in the reoperations. Recurrent IE was the indication for reoperation in 5 of 8 cases, with reoperations being performed, respectively, at 1, 10, 13, 23, and 85 months after the primary operation. Overall, the probability of freedom from reoperation at 1 and 5 years was 96.3% and 93.2%, respectively. Although only a single patient underwent reoperative surgery early (<6 months) after the primary operation, the probability of reoperation over time was significantly different in relation to the recurrence of infection or not (log-rank *p* < 0.0001). In fact, patients reoperated for recurrent MV IE were treated significantly earlier than the remaining.

## 4. Discussion

Minimally invasive cardiac surgery has consistently evolved during the last few years. To date, also complex valve diseases and high-risk subgroups of patients, such as those with IE, are routinely enrolled for operations using a right mini-thoracotomy approach in experienced heart valve centers [11,15,16].

According to current guidelines, early surgery for IE is essentially recommended in case of valve defects causing hemodynamic impairment, i.e., severe regurgitation, in the presence of uncontrolled infection, and to prevent embolic events [7,8]. In particular, early surgery has been shown to be able to avoid the progression of heart failure, prevent irreversible structural valve damage that reduces or precludes the chances of successful repair on the MV, and reduce the risks of IE recurrence and overall postoperative adverse events [17,18].

However, evidence in the literature highlights how, to date, strict adherence to the recommendations is poorly applied in real-world clinical scenarios, and only one out of four patients with IE and a potential surgical indication is actually addressed surgically or evaluated for operation [7,19]. This low rate of referral to surgery can be partially explained when considering the poor prognosis in this kind of surgical subpopulation. Despite advances in prevention strategies, antimicrobial therapies, and surgical management, IE still remains associated with substantial mortality, which ranges between 10% and 20%, even in recent reports from high-volume centers [7,20,21,22].

The right mini-thoracotomy approach is able to minimize the surgical trauma, reduce the risk of re-entry injuries in case of IE and previous cardiac surgeries, accelerate patients’ recovery, and reduce the rate of wound and other hospital-related infections in this subgroup of high-risk patients [3,4,23]. A recent randomized trial comparing MV repair via mini-thoracotomy vs. full sternotomy failed to outline an improved functional outcome at 12 weeks with the minimally invasive approach [24]. However, this study refers to very low-risk patients and excludes, among others, IE. Also, functional capacity was significantly improved at 6 weeks, suggesting that sicker patients may possibly benefit from reduced surgical trauma even at a more prolonged time interval from surgery [25]. After a15-year experience, the right mini-thoracotomy approach has become the standard of care for treating mitral and tricuspid IE in selected patients at our center. We have previously reported worthy results in terms of the feasibility and safety of the minimally invasive setting in the case of MV IE [10]. This present study on nearly 100 high-risk patients (mean logistic EuroSCORE 14.5%, almost 30% of redo procedures, and nearly 20% of patients with preoperative IE-related complications, namely, embolic events or heart failure) highlights good early and long-term outcomes with 4.4% 30-day mortality, a 1% of stroke rate, and a median intensive care unit and postoperative hospital length of stay of 1 and 7 days, respectively. Our mortality rates and postoperative outcomes are lower than estimated with the EuroSCORE and, more importantly, below recently reported data in the current literature. A systematic review on minimally invasive surgery for MV IE reports an average in-hospital mortality of 9.4%, a stroke rate of 2.3%, and an average length of hospital stay of 21.6 days, whereas reoperation during follow up was required in 9.3% of cases [16,26]. 

At follow up, reoperation was dictated by recurrent IE in 5 (5.4%) cases, but only 1 patient was reoperated on very early, 2 months after the primary operation, indicating that the eradication of the primary infection was carried out in virtually all patients (91/92, 98.9%). Consequently, in carefully selected patients, the minimally invasive approach does not jeopardize the essence of surgical treatment of IE, which imposes radical excision of all infected tissues. In this series, the etiology of IE was most commonly related to Gram-positive bacteria, with a substantially equal prevalence of staphylococci and streptococci accounting for almost 50% of the population and a lower prevalence of enterococci. The pathogen could not be isolated and sterile cultures were observed in 21 (22.8%) patients, most likely indicating sensitivity to first-line blind antibiotic therapy and, possibly, less virulent agents. The proportion of patients with negative cultures was significant, but about 50% than observed in the EURO-ENDO study, which outlined this finding in 44% of surgical candidates with IE [27]. Consequently, a potential bias against patient selection for minimally invasive MV surgery might be hypothesized and eventually correlate with the favorable outcomes observed. Actually, staphylococcal IE, which typically presents with more severe sepsis and valve destruction, represented only one-quarter of the population. Conversely, the prevalence of streptococcal culture-negative IE and the related lower virulence may correlate with some probability of indicating MV repair in a subgroup of patients with native IE.

We were able to address the majority of patients with MV IE, including reoperations in nearly 30% of the cases and mitral annular calcification in one patient. Although technically challenging and in spite of alternative hybrid strategies to mitigate anatomical risks, we do not consider annular calcification an absolute contraindication per se for minithoracotomy, irrespective of IE [14,28,29]. Admittedly, however, the minimally invasive approach was not employed in all cases to treat IE during the study period since patients suspected of more extensive IE or operated on with an emergent or salvage indication, most commonly cardiogenic or septic shock, were addressed with conventional sternotomy. In parallel, the timing of the operation is of utmost importance [30,31,32]. Despite the somewhat more aggressive timing now recommended by the current guidelines, fixed indications are lacking. The severity of valve dysfunction, the ongoing infection, and the hazards (or consequences) of embolization should always be balanced with the benefits of immediate surgery. Patients with systemic embolization represent a special challenge with respect to optimal timing. Embolic stroke, in particular, may be a true dilemma in relation to the hazards of heparinization and cerebral hemorrhage. In this respect, however, operations on cardiopulmonary bypass have been carried out with a relatively low risk of hemorrhagic infarction, anecdotally comprising conditions requiring deep hypothermia and circulatory arrest, such as aortic dissection complicated by cerebral malperfusion [33,34,35]. Thus, even in the case of cerebral embolization, IE should be managed as early as possible, reserving a “wait and see” approach to patients with very large brain infarcts, ongoing intracranial hemorrhage, and a very compromised cognitive or near comatose state. Conversely, in the case of embolization to the visceral vessels, the most commonly involved site is the splenic territory. In the presence of very large and recent infarcts, prophylactic splenectomy, although not indicated per se, may be associated with surgery for IE. In this series, this scenario occurred in two cases. Moreover, although the minimally invasive approach has been criticized with respect to the potential hazards of embolization, and subtle cerebral emboli may be detected with high-resolution neuroimaging techniques [36], the presence of IE does not appear a clinically relevant additional risk factor in our experience. In fact, postoperative stroke was observed in a single case, indicating a very low incidence, most likely unrelated to IE.

On the contrary, however, the uniqueness of the IE state is characterized by the derangement determined by the systemic infection itself, which includes acute or subacute cardiac dysfunction, a prolonged inflammatory state, and additional toxicity related to antibiotic therapy with a consequently reduced multiorgan, especially hepatic and renal, functional reserve. In this scenario, it might be speculated that a minimally invasive approach, which may inherently reduce not only surgical trauma but also additional activation of systemic inflammatory pathways, could possibly mitigate the hazards of intraoperative end-organ damage. In fact, despite this aspect being beyond the scope of this report and the sickest patients being excluded, the incidence of acute kidney injury necessitating postoperative dialysis was surprisingly low in this series (5/92, 5.5%). This could be explained by the relatively early timing of the operation in most of the cases and by the near routine use of hemofiltration during cardiopulmonary bypass. Moreover, particularly in the case of reoperations, similar considerations may apply to the unnecessary and possibly harmful dissection of previous pericardial adhesions to minimize inflammation caused by surgical manipulation. In this respect, the usefulness of the endoaortic balloon for clamping and filtration of the cardioplegic solution to avoid bicaval dissection and snaring cannot be overemphasized.

The favorable results reported in this present study outline how our current practice, based on long-standing experience with different and evolving modalities of minimal access settings for MV surgery, allows us to reach a satisfactory outcome while guiding the patient toward the safest approach. A careful selection of the patient before skin incision is required; transesophageal echocardiography is a mandatory tool to estimate the risk of embolization in relation to the size and, even more importantly, the mobility of the vegetations characterizing the IE process, to exclude aortic valve involvement, to define the likelihood of a conservative MV procedure, to exclude the involvement of perivalvular structures and abscesses, and to assess prosthetic valve impairment [37]. Angiography, vascular computerized tomography scan, or both are mandatory for the diagnosis of peripheral vascular disease.

Despite encouraging outcomes reported in the literature and recognized advantages, minimally invasive surgery for IE is currently still not widely adopted. Concerns have been expressed in relation to the challenges of the surgical learning curve, which may compromise the effectiveness of the procedure or even the ability to achieve adequate valve repair when feasible. This should definitely be taken into consideration when approaching more complex valve diseases in a program of minimally invasive surgery [38]. Thus, we recommend addressing increasingly demanding patients after the first phase of the learning curve to reach the encouraging results reported in the current literature.

## 5. Limitations

Apart from the inherent limitations of a retrospective observational study, the main drawbacks of our study are essentially the relatively small sample size and the lack of a comparison group. However, a comparative study with a standard sternotomy cohort at a single center with an advanced program of minimally invasive MV cardiac surgery is difficult, if not impossible, to realize because virtually all patients selected for the sternotomy approach present concomitant disease, e.g., in this clinical scenario, associated aortic valve IE or coronary artery disease. Consequently, even a comparison with a historical group would be biased by changes in perioperative management and surgical experience occurring from one time period to another and, thus, of little value. Similarly, EuroSCORE I was reported because EuroSCORE II was unavailable in earlier records, leading to a possible overestimation of operative mortality risk. Moreover, results refer to a high-volume and experienced center in minimally invasive MV surgery and may not be generalizable. However, the majority of patients undergoing MV surgery for IE require valve replacement, which is technically straightforward.

## 6. Conclusions

This present study depicts encouraging early and long-term results in higher-risk patients undergoing minimally invasive MV surgery for IE. Mandatory key points for a successful outcome are thorough echocardiographic assessment and peripheral vascular screening to optimize the selection of patients for the mini-thoracotomy approach and the timing of the operation to eradicate IE in the most favorable risk-to-benefit condition.

## Figures and Tables

**Figure 1 medicina-59-01435-f001:**
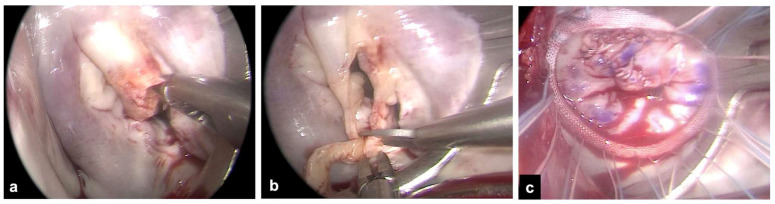
Exposure of the mitral valve via a right mini-thoracotomy and endoscopic view. (**a**) Example of large vegetation on scallop A2 with leaflet perforation. (**b**) Extensive resection of the anterior leaflet around the infected tissue. (**c**) Mitral valve repair with patch on the anterior leaflet and ring annuloplasty.

**Figure 2 medicina-59-01435-f002:**
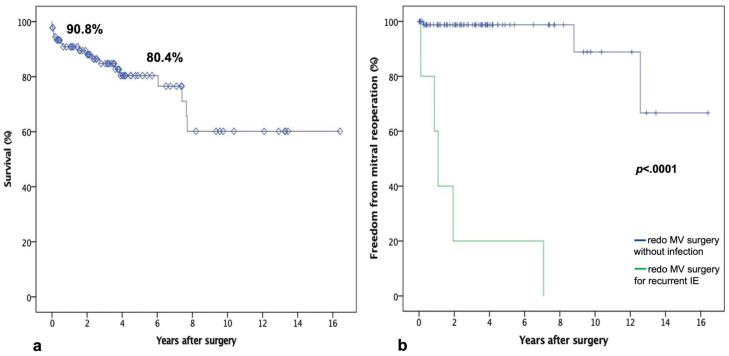
(**a**) Kaplan–Meier curve of survival probability. (**b**) Freedom from MV reoperation according to diagnosis.

**Table 1 medicina-59-01435-t001:** Baseline patients’ characteristics (n = 92).

Age (yrs)	61.1 ± 12.7
Female	34 (37%)
BMI (Kg/m^2^)	24.6 ± 5.4
Hypertension	44 (48%)
Diabetes	18 (20%)
Atrial fibrillation	21 (23%)
COPD	7 (7.6%)
Chronic kidney disease	12 (13%)
Peripheral vascular disease	2 (2.2%)
NYHA class ≥ 3	20 (22%)
IE native	80 (87%)
prosthetic	12 (13%)
Mitral annular calcification	1 (1%)
Heart failure	5 (5.4%)
Cerebral embolization	13 (14%)
Redo	26 (28%)
Ejection fraction (%)	59 ± 9
PAPs (mmHg)	38 ± 14
Log EuroSCORE (%)	15 ± 16
Early surgery	69 (75%)

BMI: body mass index; COPD: chronic obstructive pulmonary disease; NYHA: New York Heart Association; IE: infective endocarditis; MAC: mitral annular calcification; PAPs: systolic pulmonary artery pressure.

**Table 2 medicina-59-01435-t002:** Bacteriological analysis (n = 92).

Streptococcus	22 (24%)
viridans	3/22
sanguis	4/22
mitis	8/22
hemoliticus	3/22
gallolyticus	4/22
Staphylococcus	23 (25%)
aureus	16/25
epidermidis	5/25
lugdunensis	2/25
Enterococcus faecalis	14 (15%)
Other	12 (13%)
Culture-negative	21 (23%)

**Table 3 medicina-59-01435-t003:** Operative data and perioperative outcomes (n = 92).

Isolated MV surgery	82 (89%)
MV repair (native)	16/80 (20%)
MV/MP replacement	76 (83%)
TV surgery	5 (5.4%)
ASD closure	5 (5.4%)
Retrograde arterial perfusion (femoral)	85 (92.4%)
Endo-aortic clamp	40 (43%)
Trans-thoracic clamp	51 (55%)
Fibrillating heart,	1 (1.1)
CPB time (mins)	143 ± 34
Cross-clamp time (mins)	105 ± 27
Conversion to sternotomy	1 (1.1%)
Aortic dissection	-
Stroke	1 (1.1%)
Acute kidney injury	5 (5.5%)
Re-exploration for bleeding	5 (5.5%)
Mechanical ventilation > 72 h	7 (7.7%)
Reintubation	4 (4.4%)
Permanent pacemaker	3 (3.3%)
ICU length-of-stay (days)	1 [1–2]
Postoperative hospital stay (days)	7 [6–11]
30-day mortality	4 (4.4%)
Follow up (months)	98.1 [36.9–164.5]

MV: mitral valve; MP: mitral prosthesis; TV: tricuspid valve; ASD: atrial septal defect; CPB: cardiopulmonary bypass; ICU: intensive care unit.

## Data Availability

Research data are available upon request.

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
