# Peer review of "Minimally Invasive Surgery: Standard of Care for Mitral Valve Endocarditis"

_medicina, 2023, doi:10.3390/medicina59081435_

Round 1
Reviewer 1 Report
First, let me congratulate the authors for sharing this extensive experience in minimally invasive MV surgery in patients with IE. This results are impressive and even better than those available in the literature, reflecting a high-degree of expertise.
As this results come from a center with a high degree of experience in minimally invasive MV and thus the results may not be generalizable to centers with a more limited experience using this approach, even in patients without IE. In my opinion this should be underlined in the limitations of the study.
It would be of special interest to know how often the authors were able to implement a minimally invasive approach as compared to sternotomy in cases of MV IE. Furthermore, as TOE seems to be a critical step for carefully selection of candidates, a description of vegetation characteristics would be highly appreciated.
Of interest, could you provide some further anatomical details of IE ? Anterior -posterior leaflet? What about mitral annulus calcification? IE related to MAC is increasingly recognised nowadays, do you consider it a limitation for this approach? If so, consider to address this issue.
Concerning the provided microbiology data, the prevalence of culture-negative IE is very high (almost 5% higher than EURO-ENDO study). It should be emphasised.
Finally, I suggest to use EuroSCORE II instead of logistic EuroSCORE as this approach provide more reliable and comparable information.
Author Response
Thank you for your kind comments and suggestions to improve our manuscript. We address the issues raised as follows:
- The non generalizability of the approach to centers with lower experience has been added to the Limitations, pointing out, however, that IE often requires valve replacement, i.e., a technically straightforward procedure;
- As already specified in the Limitations, a control sternotomy group is unavailable because the vast majority of patients underwent minithoracotomy MV surgery during the last 10-15 years. A description of vegetations' morphology and distribution was inconstantly detailed and thus not useful. This has been specified in the text.
- We added the fact that a MAC was encountered in 1 patient and that we do not consider MACs an absolute contraindication for MICS.
- Actually the EURO-ENDO study outlines a proportion of patients with negative cultures of 44%, when analyzing the subpopulation of patients undergoing surgery, almost double than our observed data. This further suggests that surgical candidates are more likely to have received prolonged antibiotic therapy. This point has been added to the Discussion.
- Log EuroSCORE I was used solely because EuroSCORE II was unavailable in less recent records. This has been added to the Limitations, also specifying that the older scoring system in more prone to overestimation of surgical risk.
Reviewer 2 Report
I commend the authors for this submission. I think this is a great topic, as minimally invasive surgery will be more emphasized in the future.
Improvements
1. This study should be compared to the median sternotomy at your institution or historically before.
2. There should be more clarity on the objective of this study. Are you showing that with these pre-operative studies, mini mitral valve surgery can be done successfully without increased risk of morbidity/mortality? I think some other information that would be helpful are ICU days, days to discharge, time to extubation, and pain score, which are the hallmark reasons for mini mitral vs sternotomy.
3. The authors will need to temper the statement that minimally invasive mitral valves are the standard of care. In addition, even the article cited by Gammie et al. stated increased strokes in mini mitral which was not seen in this study. I would also be good to mention the recent article in Jama stating that mini mitral repair did not increase physical activity at 6wks compared to median sternotomy.
Author Response
Thank you for your kind comments and suggestions to improve our manuscript. We address the issues raised in the following points:
- As already specified in the Limitations, a control sternotomy group is unavailable because the vast majority of patients underwent MV surgery via minithoracotomy during the last 10-15 years. Thus, a comparison would theoretically be possible only with much less recent experience and a different surgical team, which would be inappropriate.
- Length of ICU and hospital stay were already reported in the text and Tables. Duration of mechanical ventilation is not reported, but the prevalence of prolonged ventilation >72 hours is included. Pain score was not available.
- MICS has become the standrad of care at out institution. We added in the Limitations that results pertain to a high-volume center with long-standing experience in MICS, specifying that results might not be generalized. We also added the recent randomized comparison between sternotomy and MICS for mitral repair published in JAMA, as suggested. The trial was underpowered for hard endpoints such as death or stroke and failed to oultine significant benefits of MICS when analyzing functional recovery with the SF-36 test 12 weeks after surgery as the primary outcome for superiority (!). Is also refers to very selected low-risk patients, excluding from randomization, among others, patients with endocarditis. Actually, a significant difference in functional recovery was outlined earlier, at 6 weeks, which may not be surprising in healthy patients, as pointed out by Dr. Maurice Enriquez-Sarano in his Invited Commentary published the same JAMA issue. These points have been added to the Discussion.